# Enantioselective Total Synthesis of Daedaleanol B from (+)-Sclareolide

**DOI:** 10.3390/molecules31010185

**Published:** 2026-01-04

**Authors:** Irene Moreno-Gutiérrez, Sonia Berenguel-Gómez, María José Cánovas-Aragón, José Luis Guil-Guerrero, Tarik Chileh-Chelh, Manuel Muñoz-Dorado, Miriam Álvarez-Corral, Ignacio Rodríguez-García

**Affiliations:** 1Organic Chemistry, CeiA3, CIAIMBITAL, University of Almería, 04120 Almería, Spain; img823@ual.es (I.M.-G.); sbg479@ual.es (S.B.-G.); mca149@ual.es (M.J.C.-A.); mdorado@ual.es (M.M.-D.); 2Food Technology Division, CeiA3, CIAMBITAL, University of Almería, 04120 Almería, Spain; jlguil@ual.es (J.L.G.-G.); chileh@hotmail.es (T.C.-C.)

**Keywords:** daedaleanol B, drimane-derived merosesquiterpenoid, *Daedalea incana*, antitumoral

## Abstract

Daedaleanol B is a drimane-derived merosesquiterpenoid isolated from the brown-rot fungus *Daedalea incana*. Herein, we report its first enantioselective total synthesis from commercially available (+)-sclareolide. A one-pot transformation afforded 11-acetoxy drimane-8α-ol, which was saponificated and selectively esterified with enantiopure L-pyroglutamic acid to give crystalline hydroxy-daedaleanol. Single-crystal X-ray diffraction of this intermediate, together with the known configuration of the chiral starting materials, enabled assignment of the absolute configuration of the daedaleanol B framework. Final elimination provided daedaleanol B, whose NMR data matched those reported for the natural product. Both hydroxy-daedaleanol and daedaleanol B exhibited time- and concentration-dependent antiproliferative effects in HT-29 colorectal cancer cells, with higher activity observed for daedaleanol B.

## 1. Introduction

Drimane-type merosesquiterpenoids (MSRDs) constitute a structurally diverse class of hybrid natural products biosynthetically derived from the combination of terpenoid and non-terpenoid building blocks, frequently amino acid-derived units. These compounds display a broad range of biological activities, including antimicrobial, anti-inflammatory, antioxidant, and antitumoral effects, and have therefore attracted considerable attention from both natural product chemists and medicinal chemists [1,2]. Representative members such as zonarol (**1**), *ent*-(+)-chromazonarol (**2**), mycoleptodiscin A (**3**), and pelorol (**4**) (Figure 1) highlight the pharmacological potential associated with this family of metabolites [3,4,5,6,7].

Wood-decaying basidiomycetes represent one of the richest natural sources of structurally original meroterpenoids. These fungi occupy chemically complex ecological niches and rely on sophisticated oxidative and detoxification mechanisms to degrade lignocellulosic biomass, a process that is frequently accompanied by the production of bioactive secondary metabolites [8,9,10,11,12].

**Figure 1 molecules-31-00185-f001:**
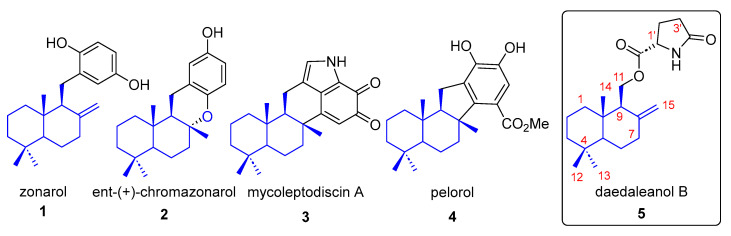
Bioactive meroterpenoids **1**–**4** and daedaleanol B (**5**).

Within this context, species of the genus *Daedalea* (Polyporales) have been reported to produce chromenes, triterpenoids, and structurally unusual sesquiterpenoid derivatives with biological relevance [13,14]. *Daedalea incana*, a brown-rot fungus widespread in tropical and subtropical forests of China, has been chemically investigated in recent years, leading to the isolation of daedaleanols A and B [15]. Phylogenetic analyses place this species within a well-defined clade of brown-rot fungi characterized by high metabolic plasticity [16].

Daedaleanol B (**5**) is a drimane-derived merosesquiterpenoid featuring an unusual hybrid architecture that combines a drimane core with an oxygenated amino acid-derived fragment. Despite its structural interest, the biological evaluation of this metabolite has been limited to a small number of cancer cell lines, where only modest activity was observed [15]. More importantly, the lack of an enantioselective total synthesis has severely restricted access to this compound, hampering deeper biological exploration and structure–activity relationship studies.

Colorectal cancer remains one of the most prevalent malignancies worldwide and a leading cause of cancer-related mortality in developed countries [17]. In this scenario, structurally novel natural product scaffolds continue to play a central role in anticancer drug discovery. Given the hybrid nature of daedaleanol B and the absence of data in colorectal cancer models, its evaluation in HT-29 cells represents a timely opportunity to reassess its pharmacological potential [18].

In this work, we report the first enantioselective total synthesis of daedaleanol B starting from commercially available (+)-sclareolide, together with its structural confirmation by X-ray diffraction and its antiproliferative evaluation against HT-29 colorectal cancer cells.

## 2. Results and Discussion

The synthetic strategy for the preparation of daedaleanol B (**5**) from (+)-sclareolide (**7**) is based on the selective esterification of the primary alcohol in drimane-8α,11-diol (**6**) with L-pyroglutamic acid, followed by a controlled elimination to install the characteristic exocyclic double bond of daedaleanol B (**5**) (Figure 1).

To ensure the robustness and generality of this approach, the key transformations were first investigated in a racemic series using (*E*,*E*)-farnesol (**8**) as a starting material. This preliminary study allowed for the optimization of the cyclization and elimination steps under conditions that avoid the use of valuable chiral substrates, while also enabling direct comparison with previously reported methodologies. In particular, our work builds on the superacid-catalyzed cyclization of farnesol described by Barrero and coworkers [19], which affords drimenol derivatives related to the albicanol framework.

In classical approaches, access to the exocyclic alkene characteristic of albicanol-type structures has relied on selenium-mediated transformations (Figure 2a). However, the use of selenium reagents under oxidative conditions presents practical and environmental drawbacks. To circumvent these limitations, we explored an alternative cyclization strategy starting from farnesyl acetate rather than farnesol (Figure 2b), inspired by earlier observations on acetate-directed polycyclizations [20,21]. Under superacidic conditions, this approach diastereoselectively furnished a polycyclic intermediate bearing a tertiary hydroxyl group, which could be exploited in a later-stage elimination to generate the desired exocyclic double bond without selenium-based reagents.

Once the feasibility of this strategy was established in the racemic series, the optimized sequence was transferred to the enantioenriched pathway starting from (+)-sclareolide. This enabled the efficient preparation of hydroxy-daedaleanol, a key intermediate that could be isolated in crystalline form and fully characterized by single-crystal X-ray diffraction prior to final elimination. The results of these studies are discussed below, beginning with the racemic synthesis from (*E*,*E*)-farnesol, followed by enantioselective synthesis from (+)-sclareolide and the biological evaluation of the final products.

**Scheme 2 molecules-31-00185-sch002:**
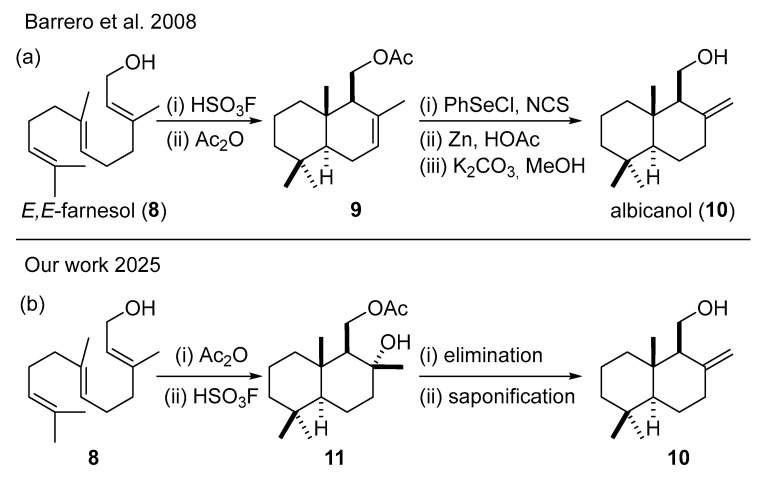
Synthesis of albicanol (**10**) from (*E*,*E*)-farnesol (**8**) (**a**) Barrero et al. [19]; (**b**) this work.

### 2.1. Racemic Synthesis from (E,E)-Farnesol

To explore an efficient methodology for the elimination of the tertiary hydroxyl group present in compound **11**, preliminary studies were carried out using the racemic material obtained from (*E*,*E*)-farnesol (**8**).

Compound **11** was obtained via superacid-promoted cyclization of farnesyl acetate using fluorosulfuric acid (HSO_3_F), affording the polycyclic drimane derivative in good yield and with high diastereoselectivity, despite the formation of up to four stereogenic centers (Figure 3). Once isolated, two different activation strategies were evaluated to promote elimination of the tertiary alcohol and generate the desired exocyclic double bond.

Activation of the hydroxyl group with methanesulfonyl chloride (MsCl), followed by base treatment (collidine), led to a mixture of alkenes with an *exo*:*endo* ratio of 62:38 (**12**:**9**). Alternatively, treatment with thionyl chloride (SOCl_2_) resulted in a slightly improved regioselectivity, affording an *exo*:*endo* ratio of 65:35. In both cases, however, the resulting alkene mixtures proved difficult to separate chromatographically, limiting their usefulness at this stage of the synthesis.

Due to the modest selectivity observed in these elimination processes, the strategy was revised to postpone alkene formation to a later stage. As shown in Figure 3, selective esterification of the primary alcohol in drimane-8α,11-diol (**6**) with L-pyroglutamic acid afforded hydroxy-daedaleanol (**13**). Subsequent elimination under optimized conditions led to a significantly improved *exo* selectivity (**5**:**14**–**77:23**), enabling chromatographic separation of daedaleanol B (**5**) and its *endo* isomer (**14**), both as mixtures of diastereoisomers as precursor **11** was a racemate.

**Scheme 3 molecules-31-00185-sch003:**
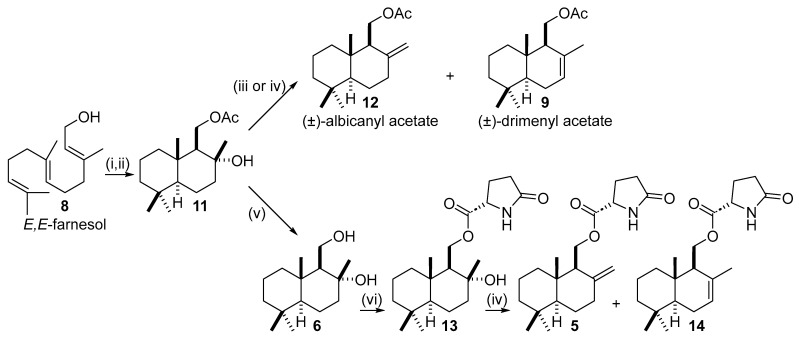
Synthesis of (±)-daedaleanol B from (*E*,*E*)-farnesol. (i) Ac_2_O, pyridine, 98%; (ii) HSO_3_F, nitropropane, 79%; (iii) SOCl_2_, pyridine, 70%; (iv) MeSO_2_Cl, 2,6-lutidine, 70%; (v) KOH, MeOH, 98%; (vi) L-pyroglutamic acid, DCC, DMAP, DCM, 70% (as mixture of diastereoisomers).

### 2.2. Enantioselective Synthesis from (+)-Sclareolide

Once the synthetic strategy had been validated in the racemic series, the enantioselective synthesis of daedaleanol B (**5**) was undertaken starting from commercially available (+)-sclareolide (**7**). This approach relied on the same key transformations established previously, namely the selective esterification of the primary alcohol with L-pyroglutamic acid followed by late-stage elimination of the tertiary hydroxyl group, while ensuring full preservation of stereochemical integrity throughout the sequence.

The synthesis commenced with the preparation of 11-acetoxydrimane-8αol (**11**) through a one-pot sequence involving nucleophilic addition of methyllithium to the lactone moiety of (+)-sclareolide, followed by a Baeyer–Villiger oxidation. According to literature precedents, treatment of (+)-sclareolide with one equivalent of methyllithium at −78 °C affords the intermediate 8α-hydroxy-12-oxo-11-homodrimane (**15**), which can be smoothly converted into the corresponding acetoxy derivative upon oxidation with in situ generated trifluoroperacetic acid [22,23].

Careful optimization of this sequence proved essential to maximize yield and suppress side-product formation (Figure 4). In particular, strict control of the methyllithium stoichiometry was required, as excess reagent rapidly led to over-addition and formation of a tertiary alcohol side product (**16**) [24]. Likewise, precise adjustment of the relative amounts of TFAA and hydrogen peroxide in the Baeyer–Villiger step was necessary to avoid competing intramolecular cyclization leading to undesired lactonic species (**17**).

Under the optimized conditions, 11-acetoxy-driman-8α-ol (**11**) was obtained reproducibly in 61% yield over the oxidation step. Subsequent saponification furnished drimane-8α,11-diol (**6**) in nearly quantitative yield (Figure 5). Selective esterification of the primary alcohol with L-pyroglutamic acid proceeded smoothly under DCC/DMAP conditions, delivering hydroxy-daedaleanol (**13**) (Figure 5).

This intermediate (**13**) proved to be a key compound in the synthetic sequence, as it could be isolated in crystalline form and subjected to single-crystal X-ray diffraction analysis (Figure 2).

The crystallographic data unambiguously confirmed the relative and absolute configuration of **13**. Importantly, because both starting materials, (+)-sclareolide and L-pyroglutamic acid have known and well-established absolute configurations, the X-ray structure of **13** provides definitive proof of the absolute stereochemistry of the synthetic pathway. This point is particularly relevant in the context of the natural product. In the original isolation study, daedaleanol B was characterized spectroscopically, but its stereochemistry was assigned only at the relative level, as no X-ray analysis was reported. In contrast, the present work establishes the absolute configuration of daedaleanol B through an enantioselective synthesis from chiral starting materials of known configuration, combined with X-ray crystallographic characterization of a direct synthetic precursor.

Final elimination of the tertiary hydroxy group in hydroxy-daedaleanol (**13**) using thionyl chloride and pyridine afforded quantitatively a mixture of *exo*- and *endo*-cyclic olefin isomers in 77:23 ratio (**5**:**14**) (Figure 5). These diastereomers were efficiently separated by semipreparative high-performance liquid chromatography (HPLC), allowing the isolation of daedaleanol B (**5**) in pure form.

The ^1^H and ^13^C NMR spectroscopic data of synthetic daedaleanol B are fully consistent with those reported for the natural product (Table 1 and Table 2).

While the literature spectra were recorded in CD_3_OD, the present characterization was carried out both in CD_3_OD and in CDCl_3_, as this last solvent provided improved signal dispersion and facilitated unambiguous signal assignment. [α]_D_ of synthetic **5** also matches literature value for the natural product [15]. Taken together, the stereochemical information derived from the enantioselective synthesis, the X-ray structure of hydroxy-daedaleanol (**13**), and the complete agreement of spectroscopic data confirm both the structure and the absolute configuration of daedaleanol B (**5**).

### 2.3. Antiproliferative Activity of **5** and **13** on HT-29 Cells

The antiproliferative activities of hydroxydaedaleanol (**13**) and daedaleanol B (**5**) were evaluated in human colorectal adenocarcinoma HT-29 cells using the standard MTT assay after 48 and 72 h of exposure. Cell viability was expressed as a percentage relative to untreated controls (Figure 3A,B). All experiments were performed in triplicate in three independent assays (n = 3), and data are reported as mean ± SD.

Both compounds induced a clear concentration- and time-dependent reduction in cell viability, with daedaleanol B (**5**) consistently showing higher activity than its hydroxylated precursor (**13**). Prolonged exposure (72 h) significantly enhanced the antiproliferative effects for both compounds.

GI_50_ values, calculated from the dose–response curves at 72 h, confirmed this trend (Figure 3B). Daedaleanol B (**5**) exhibited a lower GI_50_ value (127.5 ± 0.9 µM) than hydroxy-daedaleanol (**13**) (160.5 ± 2.1 µM), indicating higher antiproliferative potency. Although the overall activity remains moderate, the increased activity observed for daedaleanol B suggests that the presence of the exocyclic double bond contributes to the biological response.

Overall, these results demonstrate that daedaleanol B (**5**) and its hydroxylated precursor (**13**) exert reproducible, time-dependent antiproliferative effects in HT-29 cells, supporting the suitability of this scaffold for further structure–activity relationship and mechanistic studies.

**Figure 3 molecules-31-00185-f003:**
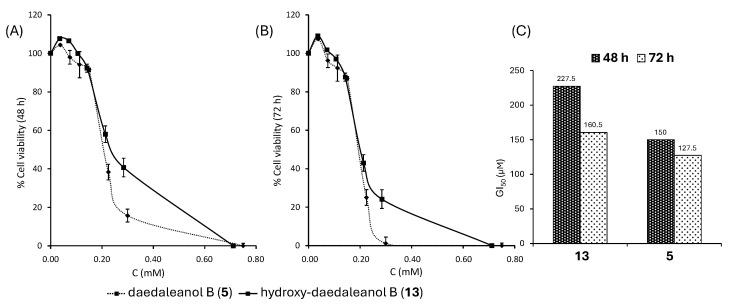
MTT Assay. Dose–response curves of HT-29 cell viability after treatment with **13** and **5** after 48 h (**A**) and 72 h (**B**), of cell exposure to the compound (mmol of compound per L of culture medium); (**C**) GI_50_ values for **13** and **5**.

## 3. Materials and Methods

### 3.1. General Experimental Details

All reactions were performed under an argon atmosphere, using oven-dried glassware in all cases. Dichloromethane was distilled from CaH_2_ under argon. THF was distilled from Na/benzophenone under argon. Dry-distilled DMF was bought from Aldrich Chemical Co. (Saint Louis, MO, USA). NMR spectra were recorded on Bruker Nanobay Avance III HD 300 MHz, Avance III HD 500 MHz, and Avance III HD 600 MHz spectrometers (Bruker Corporation, Billerica, MA, USA). Proton-decoupled ^13^C NMR and DEPT-135 were measured in all cases. When required, NOE 1D, COSY, HSQC, and HMBC experiments were used for signal assignation. Chemical shifts (δ) are reported using CDCl_3_ as an internal reference and expressed in ppm. Coupling constants (*J*) are in hertz (Hz). IR Spectra were recorded with a Bruker Alpha spectrometer. Circular dichroism was measured on a JASCO P-1030 polarimeter (JASCO Corporation, Tokyo, Japan) operating at the sodium D line. Optical rotation was measured in a Zuzi digital polarimeter model 418 with a 10 cm cell. Analytical TLC was performed on 0.2 mm DC-Fertigfolien Alugram^®^ Xtra Sil G/UV254 silica gel plates (MACHEREY-NAGEL GmbH & Co. KG, Düren, Germany). The TLC plates were visualized with UV light and 7% phosphomolybdic acid or KMnO_4_ in water/heat. Flash chromatography was performed on silicagel 60 (0.04–0.06 mm). An Agilent 1100 system equipped with a DAD detector (Agilent Technologies, Inc., Santa Clara, CA, USA), measuring at 210 nm, and a Schalau Hyperchrome 250 × 10 mm HPLC column were used for semipreparative HPLC separations. Circular dichroism (CD) spectra were recorded on a JASCO J-815 spectropolarimeter equipped with temperature control. Measurements were carried out in the 190–600 nm range using a xenon lamp as the light source and HPLC-grade methanol for instrument calibration. Samples were prepared in methanol and placed in a 1 cm path-length quartz cell. Spectra were recorded at 25 °C, and at least three scans were averaged for each sample. Data processing, including baseline correction, smoothing, and analysis of spectral maxima and minima, was performed using Spectra Manager™ Suite software. X ray diffraction investigation of compound (**21**) was conducted using an automatic diffractometer Bruker APEX-II CCD (Bruker Corporation, Billerica, MA, USA) (MoKα radiation, ω- and φ-scanning). Empirical absorption correction and systematic error correction were performed using Olex2 program. The structure was deciphered by direct methods. All calculations were performed using the SHELXS-97 software package.

### 3.2. Synthetic Methodology

#### 3.2.1. Synthesis of (E,E)-Farnesyl Acetate

Acetic anhydride (2.06 mL, 21.80 mmol, 5 eq) is added dropwise at 0 °C to a solution of (*E*,*E*)-farnesol **8** (1.08 mL, 4.32 mmol, 1 eq) in pyridine (3.33 mL, 41.25 mmol, 9.5 eq). Subsequently, a calcium chloride tower is placed, allowed to warm to room temperature, and the reaction is monitored by thin-layer chromatography (TLC). After 1 h, cold water (20 mL) is added to the mixture, and it is extracted with AcOEt (3 × 20 mL). The combined organic phases are washed with 10% HCl (40 mL), a saturated NaHCO_3_ solution (40 mL), and brine (40 mL). Then, they are dried with MgSO_4_, filtered, and the solvent is evaporated under reduced pressure. The product is purified by column chromatography, using a mixture of hexane/AcOEt (9:1) as eluent. (*E*,*E*)-Farnesyl acetate is obtained as a yellowish, clear liquid (1120 mg, 4.24 mmol, 98%).

IR ν_max_ 2980, 2917, 2855, 1739, 1443, 1379, 1227, 1021, 954, 833, 607 cm^−1^.

^1^H RMN (300 MHz, CDCl_3_) δ 5.36 (td, *J* = 7.1, 1.2 Hz, 1H, H2), 5.17–5.05 (m, 2H, H6, H10), 4.61 (d, *J* = 7.1 Hz, 2H, H1) 2.16–2.08 (m, 5H), 2.07 (s, 3H, OAc) 2.05–1.95 (m, 3H) 1.73 (s, 3H), 1.70 (s, 3H), 1.62 (s, 6H).

^13^C RMN (75 MHz, CDCl_3_) δ 171.1 (C, CO), 142.3 (C), 135.5 (C), 131.3 (C), 124.3 (CH), 123.6 (CH), 118.3 (CH), 61.4 (CH_2_, C1), 39.7 (CH_2_), 39.5 (CH_2_), 26.7 (CH_2_), 26.2 (CH_2_), 25.7 (CH_3_), 21.1 (CH_3_), 17.7 (CH_3_), 16.5 (CH_3_), 16.0 (CH_3_).

HRMS (ESI/Q-TOF) *m*/*z*: [M + H]^+^ calcd for C_17_H_29_O_2_^+^ 265.21621; found: 265.21634.

#### 3.2.2. Synthesis of 11-Acetoxy-Driman-8α-ol (**11**) from (*E*,*E*)-Farnesyl Acetate

To a solution of (*E*,*E*)-farnesyl acetate (156 mg, 0.59 mmol, 1 eq) in nitropropane (1 mL), a solution of HSO_3_F (0.36 mL, 6.25 mmol, 10.6 eq) in nitropropane (3 mL) at −78 °C and under inert atmosphere is slowly added. After 30 min, a solution of Et_3_N (1.25 mL) in hexane (1.95 mL) is added to the reaction mixture. Next, the mixture is diluted with water (20 mL) and extracted with AcOEt (2 × 30 mL). The combined organic phases are washed with water until neutral pH (40 mL), dried with MgSO_4_, filtered, and the solvent is evaporated under reduced pressure. The product is purified by column chromatography, using a hexane/AcOEt (9:1-7:3) gradient. 11-Acetoxydriman-8α-ol (**11**) (131 mg, 0.47 mmol, 79%) is obtained as a clear liquid.

IR ν_max_ 3450, 2924, 2869, 1735, 1465, 1387, 1366, 1238, 1071, 1028, 968, 939 cm^−1^

^1^H RMN (500 MHz, CDCl_3_) δ 4.34 (dd, *J* = 11.8, 4.2 Hz, 1H, H11a), 4.22 (dd, *J* = 11.8, 5.4 Hz, 1H, H11b), 2.55 (bs, 1H, OH), 2.03 (s, 3H, OAc), 1.87 (dt, *J* = 12.7, 3.3 Hz, 1H), 1.66–1.62 (m, 2H), 1.58 (dt, *J* = 13.5, 3.8 Hz, 1H), 1.52–1.36 (m, 4H), 1.27–1.19 (m, 2H), 1.16 (s, 3H, H12), 1.03 (td, *J* = 12.6, 3.6 Hz, 1H), 0.94 (dd, *J* = 12.2, 2.3 Hz, 1H), 0.87 (s, 3H, H15), 0.85 (s, 3H), 0.79 (s, 3H).

^13^C RMN (125 MHz, CDCl_3_) δ 171.4 (C, CO), 72.6 (C, C8), 62.5 (CH_2_, C11), 59.9 (CH), 55.7 (CH), 43.9 (CH_2_), 41.7 (CH_2_), 39.7 (CH_2_), 38.1 (C), 33.4 (CH_3_), 33.2 (C), 24.5 (CH_3_), 21.5 (CH_3_), 21.3 (CH_3_), 20.3 (CH_2_), 18.4 (CH_2_), 15.8 (CH_3_).

HRMS (ESI/Q-TOF) *m*/*z*: [M + H]^+^ calcd for C_17_H_31_O_3_^+^ 283.22677; found: 283.22656.

#### 3.2.3. Synthesis of (±)-Albicanyl Acetate and (±)-Drimenyl Acetate (**9** and **12**)

To a solution of 11-acetoxydriman-8α-ol (**11**) (236 mg, 0.84 mmol, 1 eq) in 2,6-lutidine (9.55 mL), MeSO_2_Cl (0.39 mL, 4.90 mmol, 5.86 eq) is added dropwise, under an inert atmosphere and at 0 °C. After 24 h, the reaction mixture is poured onto ice and extracted with AcOEt (40 mL). The organic phase is washed with 10% HCl (3 × 20 mL), with a saturated NaHCO_3_ solution (20 mL), and brine (20 mL). Finally, it is dried with MgSO_4,_ and the solvent is evaporated under reduced pressure. It is purified by column chromatography, using a hexane:AcOEt gradient (10:1-9:1) as eluent. A yellow oil is obtained from an inseparable mixture of (±)-albicanyl acetate (**12**) and (±)-drimenyl acetate (**9**) (169 mg, 0.64 mmol, 76%) in a ratio of 6.2 to 3.8, respectively.

To a stirred solution of 11-acetoxydriman-8α-ol (**11**) (275 mg, 0.98 mmol) in dry DCM (5 mL) at −78 °C, SOCl_2_ (0.108 mL, 1.47 mmol, 2.5 eq) and pyridine (0.4 mL, 4.9 mmol, 5 eq) are sequentially added, and the reaction mixture is stirred at this temperature for 10 min. Then, the reaction is quenched with NaHCO_3_ aqueous saturated solution (15 mL) and extracted with AcOEt (3 × 15 mL). Then, the organic phases are washed with brine (1 × 30 mL), dried over MgSO4, filtered, and the solvent is removed under reduced pressure. The reaction mixture is purified by column chromatography using hexane:AcOEt gradient (10:1-9:1) as eluent. A yellow oil is obtained from an inseparable mixture of (±)-albicanyl acetate **12** and (±)-drimenyl acetate **9** (202 mg, 0.76 mmol, 78%) in a ratio of 6.5 to 3.5, respectively.

#### 3.2.4. Synthesis of 11-Acetoxy-Driman-8α-ol (**11**) from Sclareolide (**7**)

To a stirred solution of (+)-sclareolide (**7**) (2.04 g, 7.98 mmol) in 20 mL dry THF, under an inert atmosphere and at −78 °C, 1.6 M MeLi (5 mL, 1 eq) is added, and the mixture is stirred for 30 min. Na_2_SO_4_ 10H_2_O (3.86 g, 1.5 eq) is then added, and the mixture is stirred at room temperature for 1 h. The mixture is then filtered to afford crude **15**, which is used in the next step without further purification. To a solution of TFAA (10.01 mL, 71.68 mmol, 15 eq) in DCM (45 mL) cooled at 0 °C in an ice bath, 30% H_2_O_2_ (3.23 mL, 32 mmol, 6.6 eq) is added. After stirring for 10 min, the crude is removed from the bath, and NaHCO_3_ (5.82 g, 69.21 mmol, 14 eq) is added in portions. After 10 min, a solution of **15** (1.2635 g, 4.74 mmol) in DCM (50 mL) is added. The reaction is monitored by thin-layer chromatography (TLC). After 30 min, the reaction is quenched with H_2_O (20 mL), and NaHCO_3_ (s) is added until pH = 7. The resulting mixture is extracted with AcOEt (3 × 50 mL), and the combined organic phases are washed with a saturated NaHCO_3_ solution (50 mL) and water (50 mL), dried with MgSO_4_, filtered, and the solvent is evaporated under reduced pressure. The product is purified by column chromatography using a mixture of hexane/AcOEt (8:2) as eluent. 11-acetoxy-driman-8α-ol (**11**) is obtained as a clear liquid (0.79 g, 2.81 mmol, 61%). Physical and spectroscopic data were in agreement with literature values [23,25].

#### 3.2.5. Synthesis of Drimane-8 α,11-Diol (**6**)

11-acetoxy-driman-8α-ol (**11**) (0.46 g, 1.62 mmol) is dissolved in KOH 2 M in MeOH (10 mL), and the mixture is stirred at room temperature until a complete reaction is observed by TLC (4 h). Then, the crude mixture is diluted with H_2_O and acidified with HCl 1 M. Then, the mixture is extracted with AcOEt (3 × 30 mL) and the reunited organic phases are washed with brine (1 × 30 mL) and dried over MgSO_4_, filtered, and the solvent is removed under reduced pressure. Drimane-8α,11-diol (**6**) is obtained as a white solid (0.38 g, 1.59 mmol, 98%).

^1^H RMN (500 MHz, CDCl_3_) δ 3.93 (m, 3H, OH, H9), 1.91 (dt, *J* = 11.8, 5.4 Hz, 1H, H7a), 1.81–1.72 (m, 1H, H2a), 1.67 (dq, *J* = 12.8, 3.2 Hz, 1H, H6a), 1.61–1.51 (m, 2H, H3a, H7b), 1.49–1.42 (m, 1H, H3b), 1.42–1.37 (m, 1H, H1a), 1.36 (s, 3H, H12), 1.31–1.22 (m, 1H, H6b), 1.22–1.15 (m, 1H, H1b), 1.12 (td, *J* = 13.3, 3.9 Hz, 1H, H2b), 0.98 (dd, *J* = 12.2, 1.8 Hz, 1H, H5), 0.89 (s, 3H, C15) 0.80 (s, 3H, C13 or C14), 0.79 (s, 3H, C13 or C14).

^13^C RMN (125 MHz, CDCl_3_) δ 75.0 (C, C8), 60.8 (CH_2_, CH_2_OH), 60.1 (CH, C9), 55.9 (CH, C5), 44.2 (CH_2_, C7), 41.7 (CH_2_, C1), 40.0 (CH_2_, C2), 37.6 (C, 10), 33.6 (CH_3_, C12), 33.3 (CH3, C15), 24.3 (CH_3_, C15), 21.6 (CH_3_, C14 or C13), 20.2 (CH_2_, C6), 18.6 (CH_2_, C3) 16.0 (CH_3_, C13 or C14).

HRMS (ESI/Q-TOF) *m*/*z*: [M + H]^+^ calcd for C_15_H_29_O_2_^+^ 241.21621; found: 241.21636.

#### 3.2.6. Synthesis of Hydroxydaedaelanol (**13**)

Under inert atmosphere at 0 °C, L-pyroglutamic acid (0.23 g, 1.8 mmol, 1.2 eq) is added to a solution of drimane-8 α,11-diol (**6**) (0.36 g, 1.5 mmol), DCC (0.38 g, 1.8 mmol, 1.2 eq) and DMAP (0.019 g, 0.15 mmol, 0.1 eq) in DCM (13 mL). Then, it is removed from the ice bath and left stirring at room temperature for 24 h, monitoring the reaction using thin-layer chromatography (TLC). Then, the reaction mixture is diluted with AcOEt (30 mL), washed with a saturated solution of NH_4_Cl (30 mL) and brine (30 mL), dried with MgSO_4,_ and the solvent is evaporated under reduced pressure. The product is purified by column chromatography using a gradient of hexane/AcOEt (1:1; 3:7; 0:1) and AcOEt/MeOH (9:1) as eluent, obtaining hydroxy-daedaleanol (**13**) as a white solid (0.37 g, 1.05 mmol, 70%).

^1^H RMN (300 MHz, CDCl_3_) δ 6.56 (bs 1H, NH), 4.49 (dd, *J* = 11.6, 3.6 Hz, 1H, H11a), 4.32 (dd, *J* = 11.6, 5.3 Hz, 1H, H11b), 4.24 (dd, *J* = 8.3, 5.1 Hz, 1H, H2′a), 2.57–2.12 (m, 5H, H2′b, H3′, H10, OH), 2.0–1.35 (m, 11H, H1, H2, H3, H5, H6, H7) 1.18 (s, 3H, H12), 0.90 (s, 3H, H13, H14), 0.88 (s, 3H, H13, H14), 0.82 (s, 3H, H15).

^13^C RMN (125 MHz, CDCl_3_) δ177.9 (C, CO), 172.0 (C, COO), 72.5 (C, C8), 63.5 (CH_2_, C11), 60.2 (CH, C9), 55.7 (CH, C5), 44.4 (CH_2_, C7), 41.6 (CH_2_, C3), 39.8 (CH_2_, C1), 38.2 (C, C10), 33.4 (C, C4), 33.2 (CH, C1′), 29.3 (CH_2_, C3′), 24.6 (CH_2_, C2′), 24.4 (CH_3_, C12), 21.5 (CH_3_, C14, C13), 20.4 (CH_2_, C6), 18.3 (CH_2_, C2), 15.9 (CH_3_, C15).

CD (c = 0.11 mg/mL, MeOH, 25.05 °C) (*θ* deg·cm^2^·dmol^−1^) (λ nm) [*θ*]_215_._5_ = +9052.1, [*θ*]_291_ = +239.9, [*θ*]_589_ = +612.6.

HRMS (ESI/Q-TOF) m/z: [M + H]^+^ calcd for C_20_H_34_NO_4_ 352.24823; found: 352.24831.

Crystal Data for C_20_H_30_NO_4_ (M = 348.45 g/mol): orthorhombic, space group P2_1_2_1_2_1_ (no. 19), a = 6.7595(7) Å, b = 7.7988(7) Å, c = 38.072(4) Å, V = 2007.0(3) Å 3, Z = 4, T = 304.90 K, μ(CuKα) = 0.639 mm^−1^, Dcalc = 1.153 g/cm ^3^, 33,728 reflections measured (4.642° ≤ 2Θ ≤ 121.666°), 3063 unique (R_int_ = 0.0817, R_sigma_ = 0.0344) which were used in all calculations. The final R_1_ was 0.0394 (I > 2σ(I)), and wR_2_ was 0.1031 (all data). CCDC 2439942 contains the supplementary crystallographic data for this paper. These data can be obtained free of charge at https://doi.org/10.5517/ccdc.csd.cc2n8lxg from the CCDC, 12 Union Road, Cambridge CB2 1EZ, UK; Fax: +44-1223-336033; E-mail: deposit@ccdc.cam.ac.uk.

#### 3.2.7. Synthesis of Daedaleanol B (**5**) from Hydroxydaedaleanol (**13**)

To a stirred solution of hydroxy-daedaleanol (**13**) (0.34 g, 0.98 mmol) in dry DCM (5 mL) at −78 °C, SOCl_2_ (0.11 mL, 1.47 mmol, 2.5 eq) and pyridine (0.4 mL, 4.9 mmol, 5 eq) are sequentially added, and the reaction mixture is stirred at this temperature for 10 min. Then, the reaction is quenched with NaHCO_3_ aqueous saturated solution (15 mL) and extracted with AcOEt (3 × 15 mL). Then, the organic phases are washed with brine (1 × 30 mL), dried over MgSO_4_, filtered, and the solvent is removed under reduced pressure. The reaction mixture is purified by column chromatography using hexane/AcOEt (7:3-1:1) as eluent. The separation of the *exo* and *endo* isomers was carried out by high-performance liquid chromatography (HPLC) using a Kromaphase 100 C18 column. The mobile phase consisted of MeOH/H_2_O (85:15, *v*/*v*), delivered at a flow rate of 2.0 mL·min^−1^. The injection volume was 100 µL, and the column temperature was maintained at 20 °C. Detection was performed using a diode array detector (DAD) at 215.4 nm. HPLC separation allowed us to obtain **5** as a white powder (0.75 mmol, 77%) and **14** as a white powder (0.23 mmol, 23%).

Spectroscopic data of **5**:

[α]^25^_D_ + 5.98 (*c* 0.14, MeOH);

IR ν_max_ 3212, 2926, 2866, 2848, 1736, 1701, 1459, 1387, 1186, 1040, 889 cm^−1^.

^1^H NMR (500 MHz, CDCl_3_) δ 6.25 (bs 1H, NH), 4.86 (s, 1H, H12a), 4.48 (d, *J* = 5.9 Hz, 1H, H12b), 4.46–4.42 (m, 1H, H11a), 4.33–4.27 (m, 1H, H11b), 4.21 (dd, *J* = 8.5, 4.9 Hz, 1H, H2′), 2.48–2.30 (m, 4H), 2.22–2.15 (m, 1H), 2.09 (d, *J* = 7.6 Hz, 1H, H9), 2.03 (dt, *J* = 13.1, 5.5 Hz, 1H), 1.77–1.71 (m, 1H, H6a), 1.70–1.63 (m, 1H), 1.60–1.49 (m, 2H), 1.46–1.39 (m, 1H), 1.35 (dd, *J* = 13.0, 4.4 Hz, 1H, H6b), 1.27–1.19 (m, 2H), 1.14 (da, *J* = 12.6 Hz, 1H, H5), 0.89 (s, 3H), 0.83 (s, 3H), 0.77 (s, 3H, H15).

^13^C RMN (125 MHz, CDCl_3_) δ 179.7 (C, CO), 172.7 (C, COO), 146.8 (C, C8), 106.3 (CH_2_, C12), 62.0 (CH_2_, C11), 55.9 (CH, C9), 55.0 (CH, C5), 54.8 (CH, C1′), 41.6 (CH_2_, C3), 38.9 (CH_2_, C1), 38.7 (C, C10), 37.3 (CH_2_, C7), 33.0 (C, C4), 28.9 (CH_2_, C3′), 24.5 (CH_2_, C2′), 23.7 (CH_2_, C6), 20.8 (CH_3_, C12, C13), 18.8 (CH_2_, C2), 14.3 (CH_3_, C15).

HRMS (ESI/Q-TOF) *m*/*z*: [M + H]^+^ calcd for C_20_H_32_NO_3_^+^ 334.23767; found: 334.23772.

Spectroscopic data of **14**:

^1^H RMN (600 MHz, CDCl_3_) δ 6.54 (d, *J* = 15.9 Hz, 1H, NH), 5.45 (d, *J* = 3 Hz, 1H, H7), 4.31 (ddd, *J* = 17.2, 11.6, 3.3 Hz, 1H, H11a), 4.15 (dt, *J* = 9.0, 4.7 Hz, 1H, H2′a), 4.10 (ddt, *J* = 11.6, 4.9, 3.7 Hz, 1H, H2′b), 2.46–2.07 (m, 5H, H3′, H1′, H11b, H9), 1.58 (s, 3H, H12), 1.54–0.96 (m, 9H, H1, H2, H3, H5, H6), 0.82 (s, 3H), 0.80 (s, 3H), 0.75 (s, 3H).

^13^C RMN (125 MHz, CDCl_3_) δ 176.9 (C, CO), 171.0 (C, COO), 130.7 (C, C8), 123.2 (CH, C7), 63.3 (CH_2_, C11), 54.6 (CH, C9), 52.5 (CH, C5), 48.8 (CH, C1′), 41.0 (CH_2_, C3), 38.6 (CH_2_, C1), 35.0 (C, C10), 32.3 (C, C4), 31.9 (CH_3_, C12) 28.3 (CH_2_, C3′) 23.7 (CH_2_, C2′), 22.5 (CH_2_, C6), 20.9 (CH_3_, C13, C14), 20.8 (CH_2_, C2), 13.6 (CH_3_, C15).

HRMS (ESI/Q-TOF) *m*/*z*: [M + H]^+^ calcd. for C_20_H_32_NO_3_^+^ 334.23767; found: 334.23774.

### 3.3. Antiproliferative Activity of **5** and **13** on HT-29 Cells

For the biological assays, the HT-29 colorectal cancer (CRC) cell line was used, supplied by the Technical Instrumentation Service of the University of Granada (Granada, Spain). This cell line is considered a pluripotent intestinal model suitable for studying various structural and molecular processes involved in cancer cell differentiation [26,27].

The assays performed included the MTT viability test (3-(4,5-dimethylthiazol-2-yl)-2,5-diphenyltetrazolium bromide) at 48 and 72 h. Experiments were conducted following the manufacturer’s instructions and the protocol described by Ortea et al. [28]. Results were obtained by measuring absorbance at 490 nm with a reference filter at 690 nm after 24 h of incubation.

## 4. Conclusions

In conclusion, we have accomplished the first enantioselective total synthesis of daedaleanol B from readily available (+)-sclareolide through a concise sequence featuring selective esterification with L-pyroglutamic acid and a late-stage elimination to install the exocyclic double bond. Isolation of crystalline hydroxy-daedaleanol enabled single-crystal X-ray diffraction, which, together with the known absolute configuration of the starting materials, supports an unambiguous assignment of the absolute stereochemistry of daedaleanol B. The synthetic compound exhibits NMR spectroscopic data fully consistent with those reported for the natural product. In HT-29 colorectal cancer cells, both hydroxy-daedaleanol and daedaleanol B display moderate, time- and dose-dependent antiproliferative activity, with enhanced effects observed for the exocyclic-alkene derivative. These results provide reliable access to daedaleanol B and a crystallographically characterized precursor, establishing a solid basis for future studies on structure–activity relationships and biological selectivity within drimane-derived merosesquiterpenoids.

## Data Availability

The original contributions presented in this study are included in the article/Appendix A. Further inquiries can be directed to the corresponding authors.

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
