# Peer review of "Enantioselective Total Synthesis of Daedaleanol B from (+)-Sclareolide"

_molecules, 2026, doi:10.3390/molecules31010185_

Round 1

Reviewer 1 Report

Comments and Suggestions for Authors

This manuscript describes the full details of the first total synthesis of (+)-sclareolide, a merosesquiterpenoid.  After the careful investigation on the construction of exo-methylene moiety by using racemic starting material prepared by the author’s tandem cyclization methodology, the target molecule was synthesized efficiently. Furthermore, the additional investigation on the antiproliferative activity of the materials synthesized is of interesting.

Chemistries described in this paper would be a nice piece of work and of interest for a number of researchers in this field.

I recommend this manuscript for publication in Molecules.

Please consider the following points:

  1. Page 4, line 121-123 and Scheme 3: Although the authors mentioned that the compound 5 was obtained in 70% yield, it must be a mixture of diastereoisomers because the starting material 11 used was racemate. It should be mentioned.
  2. Page 5, Scheme 4: An oxygen atom is lacked in the structure of compound 17. It should be a hemiacetal derivative.
  3. Page 5, Scheme 5 and page 6, line 178: The yield of 14 or the ratio of 5 and 14 should be described in the Scheme 5 or text.
  4. Page 11, line 399: Does the compound number 1 mean 5?
  5. Page 11, line 399: Although the yield of the desired product is shown as 77%, it was 70% in the Scheme 5 and text. Which data is correct?

Reviewer 2 Report

Comments and Suggestions for Authors

The present manuscript reports the enantioselective synthesis of a merosesquiterpenoid Daedaleanol B from a commercially available precursor. Key synthetic transformation was preliminary studied on the example of a racemic substrate to prove its applicability. NMR spectra of the synthetic product coincided with the spectra of natural Daedaleanol B, and absolute configuration of the product was unambiguously established basing on X-ray study of a synthetic intermediate. The synthetic procedures are described in details and the compounds are, generally, characterized with a proper set of methods.    

The work is of interest as an example of the enantioselective synthesis of natural compound and can be accepted for publication, taking into account the following remarks:

  1. The main issue is: are the absolute configurations of the synthesized product and of natural Daedaleanol B the same? As the specific rotation ([α]) is described for natural compound (ref.13), it should be compared with specific rotation of the obtained sample.
  2. What are the references [Shan2015, Matsuda2016] (line 32)? They are absent in References section.
  3. For clarity, it would be better to image configuration of both asymmetric centers in five-membered ring of sclareolide (7) throughout the text.
  4. Scheme 2, compound 11 – absolute configuration should not be given for C-OH.
  5. Yields given in Scheme 2 and Section 3.2 are different.
  6. Detailed description of standard MTT test results (section 2.3) seems excessive. On the other hand, the number of replicates and standard deviation value for GI50 should be given.
  7. In Section 3.3 caspase-3 determination and LDH analysis are mentioned. Were they performed or just mentioned accidentally?
  8. “RMN” should be corrected for “NMR” everywhere in section 3.2; multiplets should be described with diapason of chemical shifts, not with the center.
